# A One-Year Prospective Follow-Up Study on the Health Profile of Hikikomori Living in Hong Kong

**DOI:** 10.3390/ijerph16040546

**Published:** 2019-02-14

**Authors:** John W. M. Yuen, Victor C. W. Wong, Wilson W. S. Tam, Ka Wing So, Wai Tong Chien

**Affiliations:** 1School of Nursing, The Hong Kong Polytechnic University, Hung Hom, Kowloon, Hong Kong; 2Department of Social Work, Hong Kong Baptist University, Kowloon Tong, Kowloon, Hong Kong; vicwong@hkbu.edu.hk; 3Alice Lee Centre for Nursing Studies, National University of Singapore, Singapore 119077, Singapore; wilson_tam@nuhs.edu.sg; 4Withdrawal Youth Service, Hong Kong Christian Service, Tsim Sha Tsui, Kowloon, Hong Kong; sokawing@hkcs.org; 5Nethersole School of Nursing, The Chinese University of Hong Kong, Shatin, New Territories, Hong Kong; wtchien@cuhk.edu.hk

**Keywords:** hikikomori, hidden youth, health, hypertension, obesity

## Abstract

*Background*: A prospective cohort study was conducted to follow-up on 104 participants on their changes of social, psychological and physical health as exposed to the hikikomori lifestyle. *Methods*: Participants were interviewed at baseline, 6 months and 12 months by administering a set of questionnaires and anthropometric measurements. *Results*: All three health domains of hikikomori were significantly improved over the follow-up period as evidenced by: (1) increased social network scores from 2.79 ± 1.80 to 3.09 ± 1.87, (2) decreased perceived stress scores from 21.18 ± 5.87 to 20.11 ± 5.79, and (3) reduced blood pressure levels from 118/75 to 115/71 and waist-to-hip ratios. Almost half of the participants have recovered from hikikomori by returning to the workforce in society; however, the health improvements were dominant in those that remained as hikikomori and were associated with the gradual swapping of exercise practices from light to moderate level strength. *Conclusions*: With intended exposure to social worker engagement, physical assessments of the cohort study triggered the social workers to encourage participants to do more exercises, which in turn enhanced their awareness of health modification towards a better health. Engagement of social workers could be considered as part of the intended exposure for all participants, which suggested social work intervention was effective in helping hikikomori recovery.

## 1. Introduction

The Japanese term “Hikikomori” (translated as 隱蔽青年 in Chinese) describes both the condition and identity of a severe form of social withdrawal. The universal definition adopts any individual who without a clear or legitimate purpose confined him/herself at home for more than 6 months, avoiding face-to-face contact with others except family or a close person, and having a ‘Status Zero’—meaning not participating in education, training, or work [1,2]. By using this definition with the target population at age 14 to 30 years, a non-governmental organization has estimated 18,500 hikikomori cases (accounted for 2.1% of its youth population) were living in the city of Hong Kong [3]. This was consistent with the local prevalence of 1.9% reported in another recent local telephone-based survey conducted among the youth population aged 12 to 29 years [4]. Given that the prevalence of 1.6% among the Japanese population aged 15 to 39 years reported in survey 2016 was underestimated because approximately 24% of the surveyed population in 2010 would be older than 39 years in 2015 and were excluded from the estimation, the newly emerged situation with younger population and severity of Hong Kong’s hikikomori problem may be comparable to that reported in Japan [5]. Our recent study has revealed that the sedentary lifestyle living of hikikomori leads to poor physical health outcomes, in particular obesity and hypertension [1]. The top solitary activities pursued by those socially-withdrawn young individuals were sedentary in nature, such as surfing the internet, chatting on-line with strangers, and sitting in a corner [3]. The protracted confinement in a bedroom would not be conducive to the usual domestic cycle of adequate and regular sleep, causing poor sleep quality [1,6]. A recent review has discussed the similarities between hikikomori status and pervasive developmental disorders, which suggested the associations of physical problems such as headaches, neck, back and muscle pains, and gastrointestinal problems with the irregular sleep-wake rhythms of hikikomori [7]. Indeed, from the bio-psychosocial point of view, as supported by empirical evidences, the harmful impacts of social withdrawal and sedentary lifestyle on both physical and mental health have long been established. A qualitative study has shown that many hikikomori were living unhappily and having low self-esteem in general [8]. Numerous studies [9,10,11] have reported that the existence of psychiatric co-morbidity was common within 1–2 years following the onset of a hikikomori life, and that the problem often became worse during the time that the individual leads such a life. In Japan, the lifetime prevalence of mental disorders in hikikomori was almost double of the non-hikikomori population, whereas the risk of mood disorders was six-times higher among hikikomori [7]. Although the etiology remains largely unknown, many researchers believed that this is a personalized phenomenon and culturally driven. The famous book ‘Shutting out the Sun—How Japan created its own lost generation’ authored by Zielenziger [12] described how youngsters who think and behave differently from the mainstreams of the “homogeneous” Japanese society were shutting up themselves to create “free space” at home to become hikikomori. Another book discussed more in depth the diverse set of youth cultures created in Japanese society [13]. From the sociological perspective, Norasakkuniit et al. [14] reviewed the factors at the individual level on how Japanese youths marginalized themselves in their own society. In fact, a hikikomori lifestyle could be ideal for many young individuals as a personal choice to withdraw from a life that they feel is stressful. However, as more and more hikikomori cases were being identified in other Asian [3,15,16] and Western countries [17,18,19], the issues of “hikikomori” and its possible health consequences have drawn attention from public health experts worldwide [9,20]. Recent breakthroughs in blood biomarker studies suggest a biological basis of the hikikomori etiology whereby oxidative stress and inflammation may be involved causing the respective behavioral and psychological traits [21]. With the first health profile of hikikomori that has recently been established and its unclear physiological mechanism, in the present study, a previously identified cohort was followed up to understand how the social, mental and physical health states of young people who were hikikomori would change during a year. Secondarily, the incidence of young individuals recovering from the hikikomori life would also be estimated.

## 2. Materials and Methods

### 2.1. Study Design and Setting

This is a prospective cohort study designed to follow-up 104 hikikomori cases at 6 months (time point 2) and 12 months (time point 3) regarding the changes in their social, psychological and physical health while exposed to the sedentary lifestyle as reported in our previous publication [1]. In the same publication, detailed inclusion and exclusion criteria as well as the health profile at baseline of the studied cohort have also been described [1]. The definition of hikikomori was having withdrawal period for 6 months and above, but due to the tight study schedule some participants who have fulfilled all recruitment criteria but exhibited persistent withdrawal for <6 months were also recruited. However, all cases were confirmed the fulfillment of the 6-month withdrawal criterion before data included in the cohort for data analysis [1]. Individuals of the studied cohort were sourced from the core hidden youth social services of Hong Kong involving a total of nine youth service teams to cover all residential districts as operated by different non-profit organizations, and recruited through the case social workers who provided regular visits (once every 1–2 months), social counseling and life planning to encourage the clients to get back to the society. The engagement of social workers was an intended action exposed to all participants. Furthermore, the participants may also interact with social workers regarding talking about the home-based mini-medical check-up (physical measurements of this study) was also intended but was facilitated in an informal rather than a professional manner, as social workers are not health professionals by themselves. Ethical approval (HSEARS20151126002) was obtained from the Human Subjects Ethics Committee of the Hong Kong Polytechnic University.

### 2.2. Procedures of the Follow-Up Interviews

Participants were approached by their case social workers to make an appointment for interview (normally on the date of home visit) about one month before the study follow-up. On each interview day, the data collector (a trained nursing student) followed the respective social worker on their home visits to the potential participants, and conducted the measurements in the absence of the social worker immediately after their consultation. The Psychotic Screening Module of the Structured Clinical Interview for DSM Disorders Axis I (SCID-I) was used to screen the participants at each time point to exclude any psychotic and associated symptoms. Participants clinically diagnosed with mood disorders at follow-ups were also excluded. Exclusion of such participants with psychosis and clinical mood disorders was because they might not be able to complete the comprehensive set of questionnaires. However, none of the participants had psychotic ineligibility, therefore this exclusion criterion had no impact in this study. And then, the participants were proceeded with the same procedure of physical measurements and completing a set of self-administered questionnaires as performed at baseline. The interview lasted around 45–60 min. A cash voucher was given to each participant at the end of each interview as a token of appreciation. The data collectors were well trained, particularly a 20-h training was provided for the semi-structured SCID-I with the use of the instrument training kit as specified by the developer. Inter-rater reliability was assessed prior to the data collection until satisfactory agreement was achieved among all data collectors [1].

### 2.3. The Instrument and Anthropometric Measurements

The instrument consisted of the same set of questionnaires used in the previous publication, where the scales’ reliability scores and interpretation methods have also been described alongside with the baseline results [1]. In brief, the socio-demographics section captured information about any changes in financial condition, smoking habits, usual daily activities pursued such as surfing the Internet, reading comics, and watching animation. Mental health was measured by using the Chinese 10-item Perceived Stress Scale (PSS-10), the Chinese Beck Depression Inventory-II (BDI-II), the Chinese State Anxiety Scale of State-Trait Anxiety Inventory (STAI-Y1), and the Chinese 10-item Perceived Stress Scale (PSS-10). The lifestyle section mainly evaluated the degrees of distortion on way of living using the Chinese Godin Leisure-Time Exercise Questionnaire (GLTEQ), the Chinese Pittsburgh sleep quality index (PSQI), the “How healthy is your diet? Questionnaire” [22]. The social health section included the SNI to measure social connectedness, and the Chinese Family Environment Scale (CFES) to assess the three key dimensions, namely cohesion, expressiveness, and conflict. Since the Dietary questionnaire and Berkman-Syme Social Network Index (SNI) questionnaire were not available in Chinese, the English versions were translated into Chinese by two bilingual professional translators using the proper ‘translation back-translation’ method. The details of translation process and quality assurance were provided in our previous publication [1]. Whilst the physical health was assessed by the Chinese SF-36 Physical Functioning Subscale (PF-10), in addition to a series of anthropometric and physical measurements, according to the methods described previously [1]. The systolic blood pressure (SBP) and diastolic BP (DBP) values were measured twice each 5–10 min apart using the automatic oscillometric blood pressure monitor (Microlife BP A200 AFIB, Espenstrasse, Switzerland), and the average value was taken. The mercury sphygmomanometer-and- stethoscope method was used to take two BP measures (each at least 5 min apart) in case of a discrepancy over 10% between the two BP readings. Blood pressure levels were interpreted according to the Seventh Report of the Joint National Committee on Prevention, Detection, Evaluation, and Treatment of High Blood Pressure (JNC7) classification, which considered SBP/DBP over 140/90 mmHg as hypertensive and individuals with SBP ranged 120–139 mmHg and DBP ranged 80–98 mmHg as prehypertensive.

After the baseline measurement (time point 1), the PSS-14 scores were compared and found to be consistently compatible after removing the 4 items (i.e., the PSS-10 version). Concerning the long instrument to be administered by the participants, the PSS-10 was adopted since the 6 month follow-up, and hence the instrument length was shortened from a total of 156 items to 152 items for easier administration.

### 2.4. Statistical Analysis

Data collected in this study were analyzed using IBM SPSS Statistics 22.0 (IBM, Armonk, New York, USA). Frequency and percentage were computed for each of the binary or categorical variables, whereas mean and standard deviation (SD) were computed for continuous variables. Together with the anthropometric variables measured in this study, the composite scores were computed for all subscales of the instrument according to the subscale scoring schemes, and expressed as mean and SD. Missing data was replaced by the last observation carry forward method. Those continuous variables were compared across the three time points using the Repeated Measures Analysis of Variance NOVA (RMANOVA) was used to examine the changes over the three time points, whereas the F values and imputed p-values were reported. Additionally, Generalized Estimating Equation (GEE) was performed as an alternative method for RMANOVA to see whether the trend is significant or not.

## 3. Results

### 3.1. The Follow-Up Cohort

From March 2017 to June 2018, the cohort of 104 hikikomori that reported with the baseline health profile (time point 1) [1] was followed up. As shown in Figure 1, within the 12-month follow-up period, 73 and 55 participants had stayed in the study with completion of the set of questionnaires in addition to all anthropometric and physical measurements at 6 months (time point 2) and 12 months (time point 3), respectively. An overall number of 53 participants (51.0%) had completed the measurements in all three time points, whereas the attrition rates were recorded as 29.8% in time point 2 and 24.7% in time point 3. Such high attritions were due to the withdrawal for follow-up, unable to fit the interview schedule of follow-ups, and loss of connection with the case social worker. Besides, a significant portion of the follow-up population has recovered from the hikikomori by returning to the workforce in society, minus of these recovered cases, 52 (71.2%) and 28 (50.9%) of participants were remained as hikikomori in time points 2 and 3, respectively (Figure 1). The demographic characteristics of the studied cohort measured in the three time points of data collection were summarized and compared in Table 1. As expected, significant differences were only observed in age and hikikomori duration, since the cohort was followed prospectively for 12 months. However, demographics were exhibited differently in the participants who were remained, recovered, and loss to follow-up. The male-to-female ratio was 1.7:1 for both recovered and loss to follow-up as compared to the 1:1 for those remained as hikikomori. The loss follow-up group was about 1 year (18.47 ± 3.51) younger than the other two subgroups. Whilst the durations of being hikikomori were 22.96 ± 27.84, 17.62 ± 22.12 and 12.43 ± 12.39 for those remained as hikikomori, recovered and loss to follow-up, respectively.

### 3.2. Improvements in the Health Profile and Lifestyle Practice

During the one-year follow-up period, the states of all three health domains were improved in the studied cohort. Four continuous outcome variables representing three different health domains, namely SNI score, PSS-10 score, DBP and waist-to-hip ratio were shown to be statistically significant across the three time points, as analyzed by the Repeated Measure ANOVA (Table 2). Socially, the social network index score (SNI) was gradually elevated (*p* < 0.001) by 5.0% in time point 2 and 10.8% in time point 3 as compared with the baseline (Table 1).

This indicated that the participants were lesser and lesser socially isolated. However, no significant change was observed in the total interpersonal support score (ISEL) and any of its four subscales. Regarding the mental health, the perceived stress levels (PSS-10 scores) of participants were significantly reduced across the three time points (Table 2). As compared with the baseline, the number of moderate stress was reduced from 74% to 67% in time point 2 while the number of severe stress was reduced from 16.3% to 12.7% in time point 3 (data not shown). A decreasing trend was also observed in the depression levels, although statistically insignificant (Table 2). Particularly, the percentage of participants with depression at moderate level or above (BDI-II score ≥ 21) was remarkably decreased from 37.4% at baseline to 31.5% at time point 2 and 30.9% at time point 3.

However, no significant change was shown in the trait anxiety scores and they remained at the moderate level during the entire follow-up period. Physically, the blood pressure levels of participants were gradually diminished across the three time points, especially for the DBP that has achieved statistical significance *p* < 0.001. Similar reducing trend was also observed for the SBP, making the reduction of SBP/DBP of 118/75 (SD = 16/10) at baseline to 117/73 (SD = 15/9) at time point 2 and 115/71 (SD = 13/9) at time point 3 (Table 2). According to the JNC7’s classification, as compared with the 15.4% (141/91; SD = 10/5 mmHg) and 31.7% (126/79; SD = 9/5 mmHg) of the baseline, the percentages (and mean BP levels) of hypertension and prehypertension at the end of one-year follow-up were 9.0% (138/85; SD = 14/7 mmHg in five participants) and 29.1% (124/76 SD = 8/7 mmHg in 16 participants), respectively. Specifically, for the single case of type 2 hypertension (171/93) reported at baseline, his blood pressure status was remarkably improved to the JNC classification of stage 1 hypertension (151/80) at time point 2 and further to prehypertension (126/87) at time point 3. None of the hypertensive cases identified in this study was found to have a positive atrial fibrillation (AFIB), and hence no immediate risk of stroke. On the other hand, statistically significant reduction (*p* = 0.024) was shown in the waist-to-hip ratios of participants throughout the follow-ups, which was consistent with the decreasing trend of the waist circumference (Table 2). The percentage of participants with waist circumference above the cut-off that suggested health risk was reduced from 26.9% at baseline to 20.5% at time point 2 and 23.6% at time point 3. However, no significant changes were identified in other measured obesity indexes including body weight, height, and BMI. Nonetheless, GEE analysis indicated that the above four significant health outcomes (namely social network, perceived stress, diastolic blood pressure, and waist-to-hip ratio) were also shown to be consistently predictable by the everyday lifestyle of hikikomori (Table 2).The daily activity records indicated that the participants spent less time staying at home (19.11 ± 4.40 at baseline vs. 18.01 ± 5.16 in time point 2 vs. 17.47 ± 4.36 in time point 3; hours per day). When staying at home, they also spent less time in activities such as eating, using computer, and reading books and comics over the three time points. Whilst no changes were observed in hours for sleeping, watching TV, idling, and facing the wall but increased mildly the hours for mobile phone or tablets (3.11 ± 5.03 at baseline vs. 4.00 ± 4.52 in time point 2 vs. 3.88 ± 3.65 in time point 3). However, as summarized in Table 3, no significant differences were identified across the three time points in the three continuous variables of lifestyle practice representing sleep quality, physical activity level and healthy eating habits. Particularly, when compared with that of 74.0% at baseline, fluctuations were observed the percentage of poor sleepers in the two follow-ups with 78.1% (increased) in time point 2 and 70.9% (decreased) in time point 3. Despite the overall weekly leisure activity scores and physical activity levels were not significantly changed among the participants, an increasing trend was observed in the frequency of moderate-intensity exercises over the three time points (from 1.38 ± 2.32 to 1.53 ± 2.82 then 1.55 ± 2.38 times per week). Particularly in time point 2, the frequency of light activity was also notably increased from 3.12 ± 3.27 to 3.73 ± 3.77 times per week. However, the participants remained in their unhealthy ‘high sugar, high fat and low vegetables and low fruits’ diet across the three time points. These results suggested that some participants might have improved at least slightly their lifestyle practice towards a better health outcome.

As mentioned above, up to half of the participants have recovered from hikikomori and returned to the workforce in society. How the conditions of health improvement of hikikomori were differed from those who have recovered? Among the significant health outcomes identified in Table 1 above, the two subgroups followed similar trends of changes across the three time points (Figure 2). More dramatic increase in SNI scores (Figure 2a) and decrease of waist circumferences (Figure 2c) and SBP levels (Figure 2d) were observed in the participants remained as hikikomori. However, the decreasing trend of PSS-10 scores in those recovered from hikikomori were more dominant than those remained. The DBP levels decreased at the same multitude across the time points (Figure 2e). Whilst the Godin exercise scores were steadily unchanged among those remained as hikikomori, but the amount of physical activities of those recovered from hikikomori were slightly increased in time point 2 then remarkably decreased in time point 3 (Figure 2f). To take a closer look into the strength of exercises being practiced by the two subgroups, in right panel of Figure 3, participants recovered from hikikomori have superseded the strenuous activities by practicing the moderate and light exercises in time point 2 while the practice of light exercises was dramatically reduced in time point 3. On the contrary, participants remained as hikikomori have gradually increased the practice of moderate exercises across the three time points while maintained in balance by adjusting the frequency of light and strenuous exercises (left panel of Figure 3). Recovery participants have spent less time staying at home but did not show to have better sleep quality or healthier eating habits than those remained as hikikomori.

## 4. Discussion

This was the first study conducted to observe the longitudinal changes in health profile of the hikikomori, particularly in a Chinese population. In the absence of direct intervention in the studied cohort, improvements were observed in all three health domains across the three time points over one-year period. Such favorable health changes were identified as significant outcomes that were predictable by the exposure of the participants during the 12 months, in terms of (1) socially with increasing social networks, both online and offline, (2) mentally with decreasing perceived stress levels, and (3) physically with reduced blood pressure levels as well as waist-to-hip ratios.

As reported in our previous study, a number of health manifestations associated with hikikomori, such as hypertension and obesity could be at least partially contributed by their sedentary lifestyle [1]. This causes the postulation of worsening the hikikomori’s health conditions by living longer with the length of such lifestyle. However, results of this longitudinal study were seemed to be opposite to such assumption, whereas certain degrees of improvement were observed in the health profile of hikikomori throughout their one-year living course. Regarding the social and mental aspects, the majority of participants were still displaying the asocial and psychological characteristics commonly observed in hikikomori after a year, although improving trends were observed in social networks and all negative emotional states [2]. In particular, the significant reduction of perceived stress was mainly associated with those recovery participants, which requires further investigations. The underlying explanations could be something related but not limited to financial burdens, family conflicts, and personal satisfaction [23,24,25]. On the other hand, more surprisingly, the prevalence of hypertension was significantly reduced from 15.4% to 9.0% in 12 months, which was below the 12.6% adult prevalence of diagnosed hypertension [26] and the 12.8% age-specific prevalence for young people aged 15–34 [27] that have been reported in local studies. Cohesively, the prevalence of prehypertension (individuals with SBP ranged 120–139 mmHg and DBP ranged 80–98 mmHg) was also dropped from 31.7% to 29.1%, although mild. Local age-matched prevalence was unavailable for comparison since pre-hypertension was rarely investigated amongst the younger populations; however, the current prevalence was below the 42.7% prevalence reported amongst the older adults at age ≥35 [28]. The combined prevalence of hypertension and prehypertension (i.e., 38.1%) was still alerting, where the risks of transiting prehypertension into hypertension and other cardiovascular complications and metabolic disorders have been well documented [5,28,29,30].

Could such health improvements be explained by any changes of the living lifestyle? In the current study, no significant lifestyle changes were observed over the study period, except for the upward trend of practicing moderate-intensity exercises. The joint guidelines of World Health Organization and International Society of Hypertension [31,32] suggested the importance of lifestyle modifications for management of hypertension, in particular weight control by means of physical activities was regarded as the most effective. This notion has agreed with the current observed profile with both reduced blood pressure levels and waist-to-hip ratios. Endurance exercise training was known to be effective in 80% of hypertensive individuals to lower both systolic and diastolic blood pressures significantly [33]. Accumulating evidence suggested effective hypertensive management required exercises at least at moderate intensity [34,35]. This supported the notion to correlation between the reduction of blood pressure and increase of moderate-intensity exercise as observed in this study. Furthermore, the reduction in diastolic blood pressure was shown to be more significant than the reduction in systolic blood pressure in the studied cohort. Hypertension occurring at younger ages are more commonly belonging to the isolated diastolic type, because an increase of systolic BP is often caused by changes of arterial stiffness that should be more frequently happened with aging but unexpected at younger ages [36,37]. Studies also indicated that psychological distress such as job strains were found to be a risk factor for hypertension that is dominant with the increase of diastolic blood pressure [38,39]. Together with the increased practice of moderate-intensity exercises, the reduction of perceived stress in the studied cohort may provide a reasonable explanation on the decreased blood pressure and hypertensive prevalence. Nonetheless, participants recovered from hikikomori were seemed to exhibit less health improvements as compared to those remained as hikikomori, except for the perceived stress levels. Exercise was identified as a coping response to reduce stress in any employees experienced a ‘bad’ day [40]. The decreasing trend of perceived stress observed in recovered hikikomori suggested they might not need exercise as the coping response, and hence the health benefits in blood pressure and obesity were not observed. Employees in high-strain occupations were also shown to be less intended to do exercise, whereas self-efficacy was identified as an important mediating factor [41]. Furthermore, job instability and context were additional risk factors contributing to psychological morbidity and health behaviors [42]. Since occupational details and other related information were not available in the current study, future study is warranted to address why there were differences in the health improvements between the two subgroups.

However, what caused the hikikomori to exercise more intensely? Current findings suggested that participants remained as hikikomori were shown to have better health improvements than those who recovered. Whereas the gradually swapping of exercise practices from light to moderate level strength (increase of exercise intensity) was only observed amongst those remained as hikikomori, which further suggested the participants were readily to adopt a healthier lifestyle even though without recovery. As mentioned in a previous publication [1], empirical physical assessments included in this study were not only beneficial in objective measurements to strengthen the evidence, but it was also found to be important to raise the interest and awareness of participants to be more concerned with their health or at least to adopt a less “hikikomori-type” lifestyle. According to the “health belief model”, which places an important emphasis on the awareness of the threat perception (risk) is the key to trigger a series of consequences that lead to the appropriate action, which was effective in predicting the health behaviors [43,44]. The adoption of less sedentary lifestyle by staying less at home and practicing more moderate-intensity exercises were clearly an action taken by the participants even though quite a large proportion of the participants still led a secluded living style yet characterized by an increasing level of activities and exercises performed at home. And this action could be at least related to two experiences: (1) The participants were informed of the outcomes immediately after the physical assessments by the nursing researcher, which triggered their awareness and they were looking forward to having the next round of follow-up assessments [45]; and (2) As described by certain social workers, they would care about the health of their clients but they were not healthcare workers, therefore, they played a role of caregiver to remind their clients to live healthier by doing more exercises. The case social workers opined that doing health assessments at the participants’ home or in social work service centers had become an attraction in terms of encouraging them to step outside their comfort zone if not safe cocoon which has de-skilled their ability and lowered their confidence to interact with others face-to-face in the non-virtual community [46,47,48]. Another unintended yet positive outcome was the action taken by the social workers to make the most of the archived health records given to each of the assessed participant as an explicit gentle reminder for doing more exercises at home and loitering around the neighborhood community so as to make improvement in their next time point of health assessment. This implies that there is room for promoting inter-disciplinary collaboration across healthcare and social care sectors not only for the sake of doing empirical research studies but also for opening up a window of opportunity for re-engaging marginalized and invisible hikikomori at a pace they find comfortable.

Given that this study adopted a prospective cohort design, no structural intervention was provided to the participants. However, all participants of this prospective cohort study were recruited through the case social workers who had been providing them ongoing counseling and psychosocial social support. Those case social workers had the primary goal to find jobs or study opportunities for their clients and get them ready to return back to the society in a gradual and voluntarily manner. In this sense, the engagement of social workers was an intended exposure for all participants of this study. The fact indicated that such social work intervention was effective, since almost half of the participants have returned to the workforce or study institutions within a year time. The role of social workers seemed to be crucial for both preferred outcomes of hikikomori recovery and health promotion during the hikikomori stage. Therefore, a qualitative study is planned to understand the views of the social workers according to their experience in dealing with clients receiving the contemporary health assessment. Furthermore, the current study and Hong Kong social workers’ approach could also be inspiring the practice of other places. According to the recent Japanese Cabinet survey [5], the hikikomori population of Hong Kong was seemed to be relatively younger in age (13–34 years vs. 15–39 years) and longer in the hikikomori duration (2.9% versus 46.9% “over 5 years”) than the Japanese population. Despite of such differences, the current methodology and questionnaires should also be compatible to be used in different places, such as France, Italy, United States, and Japan for addressing the comparative and global health concerns of this particular field of study. On the other hand, recovered participants of current study might alike the ‘affinity group’ designated by Tajan et al. [5], in order to provide further information for exploring the triggering factors.

There are, however, several limitations in this study. Because of the hidden nature of the target participants who are one of the hardest groups for engagement and rapport building following their protracted period of seclusion at home, subject recruitment is considered as the most difficult part of the study. It caused the small sample size and high attrition rate as a major limitation. Although the sample size was sufficiently enough to achieve statistical significance when certain measured variables were compared, the high attrition rate due to loss of follow-up would contribute to selection bias that threatening to the internal validity [49]. Particularly in this study, participants intended to loss to follow-up could be those who were more hidden and living with more hermetic and sedentary behavior that causing inaccurate interpretations. They were shown to be younger at age and shorter in hikikomori duration, which requires more in-depth investigations. Certain stratification-based techniques could be used to correct for such selection bias in future analysis. Furthermore, although participants of this study were recruited from multiple centers, sampling through a single agent i.e., social work is also considered as a major limitation because many hidden cases still could not be reached and sampled. It is suggested that in future studies, other agencies such as secondary schools, student residency of universities, family-based services, medical units, and relevant online forums can also be approached for sampling.

## 5. Conclusions

The hikikomori lifestyle which is largely sedentary in nature that could be a risk behavior, but a longitudinal study has observed improvements in all three domains of the health profile, namely social networks, perceived stress, and blood pressure levels (especially the diastolic blood pressure). The reduction of blood pressure levels and prevalence were consistent with the reduction of waist-to-hip ratios as well as the increase of moderate-intensity exercise over the follow-up period. Whilst the reduction of perceived stress was more specifically associated with the participants recovering from hikikomori, physical assessments followed by encouragement from social workers to do more exercises might enhance awareness of hikikomori youths in health modification towards a better health. There is implication for promoting inter-disciplinary collaboration across healthcare and social care sectors for conducting further empirical studies and delivering engagement interventions at a pace that secluded and marginalized hikikomori find comfortable.

## Figures and Tables

**Figure 1 ijerph-16-00546-f001:**
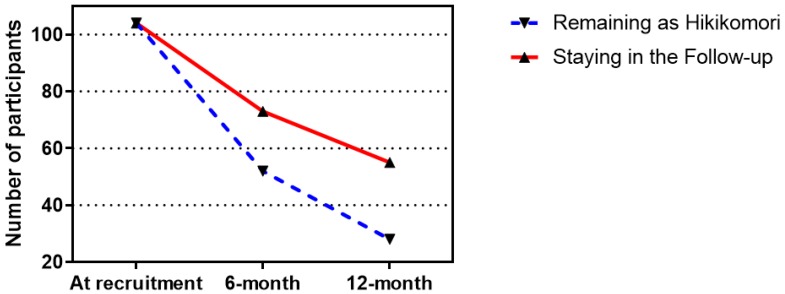
Number of participants at each time point.

**Figure 2 ijerph-16-00546-f002:**
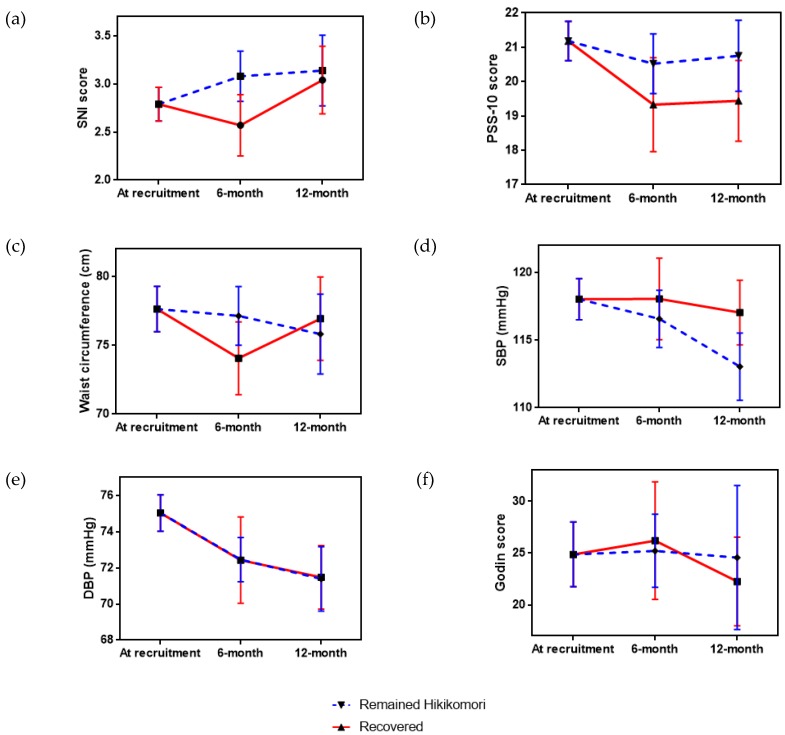
Comparison of the pattern changes among significant variables (**a**) SNI score; (**b**) PSS-10 score; (**c**) waist circumference; (**d**) SBP; (**e**) DBP; (**f**) Godlin score between the participants remained as hikikomori and those recovered from hikikomori. SNI: Social Network Index; PSS-10: 10-item Perceived Stress Scale; SBP: systolic blood pressure; DBP: diastolic blood pressure.

**Figure 3 ijerph-16-00546-f003:**
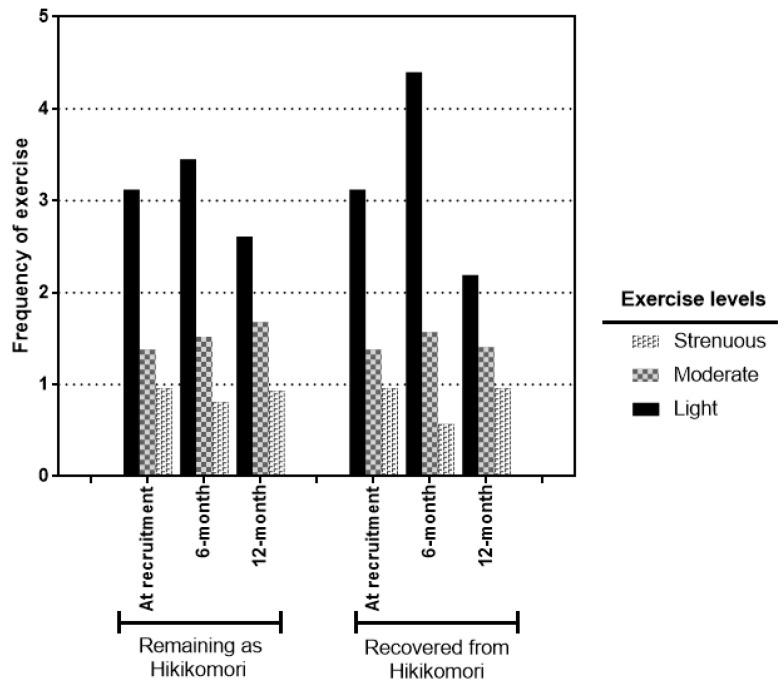
The frequencies of different intensity levels of exercise practiced by participants remained as hikikomori and those recovered.

**Table 1 ijerph-16-00546-t001:** Demographic characteristics of the hikikomori youths followed in the 3 time points.

Variables		Time Point 1 (*n* = 104)	Time Point 2 (*n* = 73)	Time Point 3 (*n* = 55)	*X*^2^ *p*-Value
		Number (Percentage)			
Gender	Male	62 (59.6)	42 (57.5)	31 (56.4)	0.916
Age (years)	Mean ± SD	19.02 ± 3.62	20.21 ± 3.50	20.84 ± 3.64	0.006 ^∆^
	Range	13–31	13–31	14–32	
	Age group 13–17	41 (39.4)	18 (26.0)	11 (20.0)	0.090
	Age group 18–24	54 (51.9)	47 (63.0)	37 (67.3)	
	Age group 25–34	9 (8.7)	8 (11.0)	7 (12.7)	
Duration of being hikikomori (Months)	Mean ± SD	16.14 ± 20.16	25.16 ± 22.73	34.96 ± 25.76	<0.001 ^∆^
	Range	3–120 ^‡^	7–127	13–132	
Living	Alone	2 (1.9)	3 (4.1)	1 (1.8)	0.874
	With immediate family	100 (96.2)	68 (93.2)	54 (98.2)	
	With relative ^†^/friends	2 (1.9)	2 (2.8)	0 (0)	
Residential	Self-owned/self-rented	2 (1.9)	2 (2.7)	1 (1.8)	0.773
	Not self-owned/Not self-rented	102 (98.1)	71 (97.3)	54 (98.2)	
Financial source	Self	23 (22.1)	25 (34.2)	21 (38.2)	0.297
	Family/relatives	80 (76.9)	47 (64.4)	34 (61.8)	
Smoker	Current	13 (12.5)	6 (8.2)	5 (9.1)	0.861
	Quitted	10 (9.6)	7 (9.6)	4 (7.3)	
	Never	81 (77.9)	60 (82.2)	46 (83.6)	

Note: ^†^ Immediate family members include parents, spouse, brothers, sisters, sons and daughters; Extended family members other than those listed above are referred as relatives. ^∆^ Determined by one-way ANOVA. *Χ^2^* = Chi-squared test. ^‡^ All cases have confirmed the fulfillment of the 6-month withdrawal criterion before data was included in the cohort for analysis [1].

**Table 2 ijerph-16-00546-t002:** The key social, psychological and physical health variables of identified in hikikomori followed up for one year.

Variables		Time Point 1	Time Point 2	Time Point 3	RMANOVA F	(Imputed) *p-*Value	GEE Beta	*p-*Value
		(*n* = 104)	(*n* = 73)	(*n* = 55)				
		Mean ± SD						
**Social ^†^ support**	SNI score (0–7) ^a^	2.79 ± 1.80	2.93 ± 2.06	3.09 ± 1.87	12.174	**<0.001**	0.275	**0.009**
	ISEL total score (0–48) ^b^	24.60 ± 6.30	24.63 ± 5.99	24.75 ± 6.89	1.074	0.302	0.269	0.493
	Appraisal (0–12) ^b^	6.81 ± 2.16	6.90 ± 1.87	7.11 ± 2.18	2.463	0.096	0.170	0.267
	Tangible (0–12) ^b^	6.20 ± 1.71	6.29 ± 1.53	6.20 ± 1.73	0.618	0.496	−0.010	0.933
	Belonging (0–12) ^b^	6.00 ± 2.45	5.89 ± 2.53	6.05 ± 2.44	2.343	0.104	0.136	0.342
	Self-esteem (0–12) ^b^	5.59 ± 2.08	5.55 ± 2.14	5.38 ± 2.39	0.544	0.577	−0.052	0.685
**Psychological ^†^**	Perceived stress (0–40) ^c^	21.18 ± 5.87	20.18 ± 6.25	20.11 ± 5.79	3.437	**0.045**	−0.709	**0.028**
	Depression (0–63) ^d^	17.17 ± 11.49	16.48 ± 10.87	15.76 ± 11.80	1.327	0.265	−0.658	0.241
	T-anxiety (20–80) ^e^	44.22 ± 12.17	46.48 ± 12.97	42.45 ± 11.16	2.105	0.131	−0.621	0.291
**Physical**	SBP (mmHg)	118 ± 16	117 ± 15	115 ± 13	0.955	0.386	−1.212	0.108
	DBP (mmHg)	75 ± 10	73 ± 9	71 ± 9	10.223	**<0.001**	−2.058	**<0.001**
	BMI (kg/m^2^) ^f^	22.28 ± 6.88	22.02 ± 5.75	22.71 ± 6.58	1.753	0.180	0.127	0.156
	Waist circumference (cm) ^g^	77.64 ± 16.87	76.26 ± 12.13	76.37 ± 15.44	2.043	0.138	−0.643	**0.050**
	Waist-to-hip ratio	0.82 ± 0.09	0.81 ± 0.75	0.81 ± 0.09	3.967	**0.024**	−0.009	**0.012**

^†^ The social and psychological scales were measured with total scores for comparison between the 3 time points of measurements. ^a^ A lower SNI score indicates more social isolation risk while ^b^ higher ISEL total and subscale scores means more social support. ^c^ Perceived stress scored 14–26 indicates moderate stress and 27–40 indicates severe stress. ^d^ Beck depression scored 17–20 = borderline clinically significant depression, 21–30 = moderate level, 31–40 = severe level, >40 = extremely severe. ^e^ T-anxiety score >40 = clinically significant level. ^f^ BMI Classification: <18.5 = Underweight, 18.5–22.9 = normal, 23–24.9 = overweight/pre-obese, ≥25 = obese. ^g^ Waist classification (Cut-off for Male ≥ 90; Female ≥ 80) as a measurement of visceral fat mass suggesting long-term health risk association with obesity. RMANOVA: Repeated Measures Analysis of Variance; GEE: Generalized Estimating Equation.

**Table 3 ijerph-16-00546-t003:** A summary of lifestyle measurements in hikikomori followed up for one year.

Variables	Time Point 1	Time Point 2	Time Point 3	RMANOVA	Imputed	GEE	*p*-Value
	Mean ± SD			F	*p*-Value	Beta	
**PSQI score**	6.57 ± 2.98	7.44 ± 6.24	6.45 ± 2.97	1.622	0.208	0.009	0.953
**Weekly Leisure Activity**	24.88 ± 31.81	25.51 ± 25.42	23.44 ± 30.31	2.731	0.081	0.826	0.638
**Healthy Eating score**	12.57 ± 4.85	12.21 ± 5.22	12.25 ± 4.98	0.200	0.813	−0.017	0.928

PSQI: Pittsburgh sleep quality index.

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
