# Peer review of "A One-Year Prospective Follow-Up Study on the Health Profile of Hikikomori Living in Hong Kong"

_ijerph, 2019, doi:10.3390/ijerph16040546_

Round 1

Reviewer 1 Report

I had a good time reading the paper entitled “A One-Year Prospective Follow-Up Study on the Health Profile of Hikikomori Living in Hong Kong”. It is a good article with innovative results about an under-investigated topic, and it fits the scope of IJERPH journal. The paper shows that, among a cohort of 104 young people living as hikikomori, half of the participants have “recovered” from hikikomori by returning to the workforce in society, and three domains of health profile of hikikomori were significantly improved over the follow-up period. Overall, it shows the effectiveness of social work intervention targeting hikikomori cases. The paper should be considered a good contribution to the field, with possible comparative studies (Japan, Italy, France, Spain, South Korea, China, etc.) in the near future, inspired by the same methodology and questionnaires. However, because I do not have an expertise in anthropometric measurements and statistical analysis another reviewer should be called to evaluate them.

To me, the paper is not far from its final version. I made some remarks and recommendations below, and I would like some aspects to be revised.

1/ Although the article is about “hikikomori”, as surprising as it may seem, the authors do not provide with a definition of hikikomori. Please state the definition at least in footnote or endnote.

2/ If I am not wrong, nothing is said about the age of the participants. Please explain why. If there’s something here that could be considered as a limitation of the research, please include it in the end of the discussion. As of now, in Japan, hikikomori individuals age is considered to be from 15 to 39 (Naikakufu and MHLW definition). Is it the same in Hong-Kong?

3/ The reader would like to know if the term hikikomori has a translation in Chinese, and if the individuals recognize themselves as such (in this case please indicate the word they use). Please provide the Chinese characters and transliteration at least in footnote or endnote. Or is it just a term used by researchers and professionals?

4/ Please also state if you use the word “hikikomori” alone to designate the individual concerned like in French, Italian, Spanish and English (“he is hikikomori, he is a hikikomori”) or if you add another word to indicate the subject or situation like in Japanese (respectively hikikomori toujisha, hikikomori joutai).

5/ Authors should write if the prevalence/estimation of hikikomori population in Hong Kong is known or not, and using which definition (including age).

6/ Although I wrote around 7 articles and a book about hikikomori in French (a second forthcoming in English, Routledge), I do not mind them not being quoted since my approach is qualitative and very different from the authors’. However, I found that some data in the one I published with my co authors in Japan Focus – The Asia Pacific journal about the Naikakufu report (Japan Cabinet office) could find a place in the current paper.

7/ Authors excluded individuals with mood disorders and individuals with psychotic symptoms. Whereas it could be considered right for some researchers, others considering hikikomori as a Modern type depression (Kato) could criticize the fact of excluding those with depressive symptoms, especially since social isolation increases depressive symptoms, and presenting depressive symptoms does not exclude someone from the hikikomori category in Japan.

8/ Overall, please contextualize your paper with the recent data of the Japanese Naikakufu report in order to contrast your data with the Japanese ones and contribute more to the international debate on the topic.

9/ In this sense, I feel studies using a similar methodology and questionnaires should be done in France, Italy, US, Japan etc in comparative/global mental health perspective. It would be a great contribution to the field.

10/ Finally, the role of social workers seems crucial, though not much is said. It should be explained further in a distinct article using a qualitative approach. If there are such improvements, such as indicated by the sentence “social work intervention was effective”, the Japanese, and not only them, would like to get inspiration from HK social workers’ approach. Please indicate if another article is forthcoming using a qualitative approach. If it is not the case, then it would be disappointing to have improvements described without knowing how to replicate them in other countries… 

Thank you for the opportunity to read and review this paper.

NT

Author Response

As attached.

Reviewer 2 Report

The article reflects an interesting empirical study on hikikomori phenomenon in Hong Kong. However, it should be improved in order to be published. In the following lines I will suggest some changes:

1) Abstract: Although in the article the conclusions are supported by the results, it does not seem so in the abstract. The fact of Social work intervention being effecttive cannot be deduced from the results summarized in the abstract.

2) Introduction: It is short and incomplete. First of all, the hikikomori phenomenon should be defined. Secondly, in page 2, lines 51-52, you say that “many researchers believe that it is a personalized phenomenon and culturally driven, but you do not clarify what it means. A reference should support this affirmation and there is no reference. I recommend the author go through the book of Michael Zielenziger, Shutting Out the Sun: How Japan Created Its Own Lost Generation. Different papers of Roger Goodman, Tukka Toivonen and Yukiko Uchida are also highly recommendable on this regard. In addition, you should say from the beginning that the study will be done in Hong Kong, because it is not clear, and including the incidence of the phenomenon in Hong Kong comparing with Japan and another countries that deserve to be highlighted.

3) Materials and methods: Regarding this section, I have nothing to say. I think that all materials, procedures, strategies, and resources used, has been correctly explained. In turn, all materials and methods are appropriated.

4) Results: In page 3, line 130 you mention certain attrition, but not why it has been produced, and it is a datum important to know. Please, che kif the percentages included in page 4, line 132 are right. In page 6, line 179, it is said tha “no significant differences were identified across the three waves in the three continous variables (…) representing sleep quality, physical activity level and healthy eating habits”. Next you detail theses results the first and the second, bat not the third.

5) Discusion and Conclusions: Nothing to say on this regard. However, the discusion should be reduced in order to enlarge the conclusions, which are quite short.

Author Response

As attached.

Reviewer 3 Report

First, I agree with the authors that this was a difficult study to conduct. I do appreciate the efforts of the researchers. I believe that it provides valuable information to health-care workers on the impact of home-visits to hikikomori people. However, the current paper needs to be more organized, and extensive revision is necessary before it suits publication.

There are three major concerns of this manuscript.

1.       The citations and references do not match the arguments or ideas presented in the introduction and discussions, raising the question of integrity.

2.       The study objectives, methods, and the discussions of results did not match with one another, raising the question of logic and questions to the study purpose.

3.       The subject of interest in this study was not clearly defined. Please make it clear if the authors wish to discuss the improvement of health/behaviors of the hikikomori people who remained hikikomori after 12 months of social work follow up? Or the progression of the health/behaviors of the hikikomori who recovered during the 12 months of social work follow up?

4.       The manuscript needs a grammar check.

Please refer to the following comments for improvement.

Comments for the abstract

1.         Is living lifestyle the intervention for this study? If yes, please state the intervention in methods.

2.         What is the main subject that the authors want to discuss in this study? The profile changes of those who remained hikikomori after 12 months follow up? Or the profile changes of those who recovered from hikikomori during the follow up?

3.         The demographics of the participants in study 1, 2 and 3 should be listed out in tables.

4.         The conclusion does not match the objective.

Comments for the introduction

1.         The introduction contains wrong citations of previous studies. For example, Koyama’s paper did not seek to provide incidence of mental disorders; the samples collected in Yong’s study was not limited to Hong Kong; and Borovoy’s article did not study the onset of hikikomori, etc. Please make sure your citations are correct for it raises the question to integrity.

2.         The use of “pandemic” for hikikomori (line 55 pg. 2) is controversial.

Comments for the methods

1.         How the authors presented their results, sounds like an intervention of social work. Methods of intervention and its goal were not mentioned in the methods. What was/were the intervention/s? What were the predicted outcomes?

2.         Use “time” or “study” instead of “wave”, pg.2 , line 66.

3.         Participants or potential participants? Pg.2, line 74

4.         What makes the participant eligible? Pg.2, line 79

5.         Who did the training for the data collectors? How were the data collectors being found competent in administering the SCID-I? What is the inter-rater reliability that the authors talk about here? This needs further illustrations. Pg. 2, lin 82-85

6.         Pg. 2-3, line 87 – 105. The reliability scores, how the scales were presented, how the authors define or interpreted the findings of the scales were not discussed. Also with the translated SNI, had it been tested? If yes, the reliability scores should be presented. Did the subject in this study complete PSS-14 or PSS-10? It sounds like the authors removed the 4 items after the PSS-14 had been administered.

7.         Pg. 3, line 106-110. If hypertension was recorded, please give the reading of hypertension defined in this paper. The hypertension information should also be presented together with the demographics.

Comments on Results and Discussions:

1.         Please provide demographic characteristics of participants in the table for study 1, 2 and 3.

2.         Pg.3-4, line 125-133. I understand that 51 dropped out in the study? At wave 2, almost 31 (29.8% of 104) dropped out, and 19 (24.7% of 73) dropped out. The number doesn’t tally. Please describe your results clearly. When you said, only 52 remained as hikikomori in wave 2, does it mean 21 had recovered in the first 6 months of intervention? The dropped out, were they out of follow-up, not in the program, or closed case? These should be mentioned explicitly in the methods.

3.         Table 1 needs further explanation. What are the lowest and highest scores of each scale? Did the authors measure with rank? Or with total scores?

4.         Pg.6, line 148, I believe that the results were insignificant instead of significant?

5.         Pg.6, line 149. this phrase, “depression at moderate level or above” come out the first time. How did the authors define the depression at a moderate level or above?

6.         Pg.6, line 153-156. The physical condition of the participants in table 1, including the dropped out, should provide information on sex and age, as these are the influential factors. If waist circumference is measured, height should be measured.

7.         Pg.6. line 162-163. What is positive AFIB?

8.         Pg. 6, line 174~ The authors wrote about lifestyle here. Lifestyle measurement should be recorded according to sex. Was the intervention about asking the participants to change their lifestyles? These were not presented in the methods. If lifestyle was instructed to change, please include them in the methods, and discuss these changes in the discussion,

9.         Pg. 7, Line 191, May I clarify that the results in Figure 2 describe the differences between the recovered hikikomori and the remained hikikomori in follow-up? If yes, the remained hikikomori seems to do better in all aspect than the recovered hikikomori. So, what is the definition of recovered hikikomori? And what good is it to recover than to remained hikikomori? The demographics of those who recovered and remained hikikomori, as well as the drop out should be provided, including all their baseline and follow up characteristics.

10.     Pg.8, line 232, Reference 15 and 18 has nothing to do with this discussion.

11.     Pg.9, line 236. The statement here was not mentioned in the results. How many participants had hypertension during baseline? And the definition of hypertension? Also, the prevalence of hypertension in which group had decreased was not specified.

12.     Line 239. Please provide the prehypertension value defined by the authors in this paper, and also present the result before discussing them.

13.     Line 240. As you had not done age-matched in your study, this phrase is just adding more confusion to your discussion.

14.     Line 246-265. As lifestyle changes had not been stated in the introduction nor methodology, the discussion here does not make sense.

15.     Line 266. The authors should be aware that not providing health promotion and education was merely the researcher’s decision, not a limitation of the cohort study, especially when you have 3 time-points of intervention.

16.     Line 267-276. The intervention by the social workers should be addressed in the introduction and methods. What I understand from this paragraph is that the objectives of this study are to provide empirical evidence to the effectiveness of social work interventions by measuring the outcomes with measured scales. Then, the authors should focus on discussing the changes seen in the participants which were related to the interventions. Instead, the authors kept focusing on discussing the changes in scales, but it had not discussed the benefits of the interventions. It is also important to know if the interventions work better with the old cases or new cases of hikikomori.

17.     Line 279-299. I would say this is the only part that matters the discussions in this paper following the introduction.  This paragraph is the only part that had discussed the possible effect and influence of social work interventions on the physical health improvement of the participants. The type of social work interventions should be clearly stated in the methods, followed by the outcome expected by these interventions. The authors should also discuss the differences between the dropout, the recovered hikikomori, and the remained hikikomori.

18.     Line 300. The authors discussed that the small sample size and high attrition rate as a major limitation. It is important for the authors to explain what are the possible biases to the results that this limitation would have caused?

19.     Line 303-304. As stated in the early comments, the statistical analysis of the participants should consider the influence of sex, age, and duration of hikikomori.

20.     Line 305. “Furthermore, although participants of this study were recruited from multiple centers, sampling through a single agent i.e. social work is also considered as a major limitation because many hidden cases still could not be reached and sampled.” This information was not available in sample recruitment methods.

Author Response

As attached.

Round 2

Reviewer 3 Report

The revised manuscript is better understood. However, there are still several concerns. 

Pg. 2, line 62. Koyama's paper indicated that 54.5% of the hikikomori cases who fulfilled diagnostic criteria of having at least one psychiatric disorder in their lifetime, and those were not hikikomori were 29.5%. The authors need to understand that Koyama's study was a cross-sectional population study, and it was not an age-matched control population study. "Incidence" refers to new cases. In Koyama's study, it only provides "lifetime prevalence". 

2.  Pg. 8, line 246. The authors asked a question themselves, " how the conditions of health improvement of hikikomori have differed from those who have recovered?" This is an interesting question who the audience would like to know as well. The results seemed to point out that those who had recovered from hikikomori had less improvement in health compared to the participants who remained in hikikomori. The authors had discussed a lot on the health improvement of those who remained in hikikomori, but had not explained why the recovered hikikomori did not do better.

3. The effect of loss to follow up bias was not discussed.

4. Pg.9, 267, and Pg. 11, line 347. a prospective cohort design has nothing to do with whether the structural intervention was provided to the participants. In fact, line 334, 352, the engagement of social workers was not natural exposure, but intended exposure, as these are the cases who were engaged by social workers. 

5. Pg.9, line 269. How the authors had presented the results for abstract, and discussions, "increasing social networks...", "decreasing perceived stress levels", "reducing blood pressure" indicated an intentional approach. If there was no direct intervention, the results were supposed to be unintentional.  

6. Table 1, duration of being hikikomori (months) range 3-120 (wave 1), 3-127 (wave 2), 3-132 (wave 3), leads to 2 questions. 

 6.1 What is the definition of hikikomori in this study? withdrawal period for 3 months and above? or 6 months and above?

 6.2 If this is a cohort study, the lower range for wave 2 and wave 3 should be more than 3 months. 

7. Please explain why the authors choose the term  "wave", instead of "time" or "study". Wave usually indicates a large number of participants of population study, often used in the unspecified cohort. When there is a small number of participants and with specified follow up is available, time and study are usually preferred. 

8. It would be more informative if the authors can provide the demographic details (age, gender, duration of being hikikomori) of those who had recovered from hikikomori, remained in hikikomori, and loss to follow-up. 

9. The scores for the tested instruments were provided in table 2. What did the scores indicate? To help the audience to relate to the meaning of the scores, it should be explained either in the methods, or described in the footnote of table 2 at least. 

Author Response

The authors have carefully considered all reviewers’ and recommendations, and replied to each reviewer
point-by-point. The major changes made are listed as follow:
1) Updated the information regarding the reference of Koyama et al.
2) The study design of prospective cohort study is reconfirmed and clarified that the engagement of social
worker and related interaction between the participants and social workers were intended exposure.
Different parts of the manuscripts were amended in line with this clarification.
3) The less health improvement observed among the recovered hikikomori was discussed.
4) The selection bias of loss to follow-up was discussed.
5) Corrected the wrong data about the duration of hikikomori in Table 1, and clarified the definition of
hikikomori in this study in the methods.
6) “Waves” were replaced with “Time points”.
7) Demographics of those remained as hikikomori, recovered cases, and loss to follow-up were added in
the results.
8) Meanings of scores in Table 2 were added in the footnote.
9) Four references are added in the revised manuscript.